# Solution-processed nanographene distributed feedback lasers

Víctor Bonal [1], Rafael Muñoz-Mármol [1], Fernando Gordillo Gámez[2], Marta Morales-Vidal [1], José M. Villalvilla [1], Pedro G. Boj [3], José A. Quintana[3], Yanwei Gu[4], Jishan Wu [4], Juan Casado [2] & María A. Díaz-García [1]

The chemical synthesis of nanographene molecules constitutes the bottom-up approach toward graphene, simultaneously providing rational chemical design, structure-property control and exploitation of their semiconducting and luminescence properties. Here, we report nanographene-based lasers from three zigzag-edged polycyclic aromatics. The devices consist of a passive polymer film hosting the nanographenes and a top-layer polymeric distributed feedback resonator. Both the active material and the laser resonator are processed from solution, key for the purpose of obtaining low-cost devices with mechanical flexibility. The prepared lasers show narrow linewidth ( < 0.13 nm) emission at different spectral regions covering a large segment of the visible spectrum, and up to the vicinity of the near-infrared. They show outstandingly long operational lifetimes (above $10^5$ pump pulses) and very low thresholds. These results represent a significant step forward in the field of graphene and broaden its versatility in low-cost devices implying light emission, such as lasers.

[1] Departamento Física Aplicada and Instituto Universitario de Materiales de Alicante, Universidad de Alicante, 03080 Alicante, Spain. [2] Department of Physical Chemistry, University of Málaga, Andalucía Tech., Campus de Teatinos s/n, 29071 Malaga, Spain. [3] Departamento Óptica, Farmacología y Anatomía and Instituto Universitario de Materiales de Alicante, Universidad de Alicante, 03080 Alicante, Spain. [4] Department of Chemistry, National University of Singapore, 3 Science Drive 3, Singapore 117543, Singapore. Correspondence and requests for materials should be addressed to J.W. (email: chmwuj@nus.edu.sg) or to J.C. (email: casado@uma.es) or to M.A.D-G. (email: maria.diaz@ua.es)

The vanishing gap in graphene is strongly detrimental for the development of electrically semiconducting applications and, for the same reason, also for the appearance of luminescence in the two-dimensional (2D) carbon-atom sheet[1]. To remediate this adverse situation, the aperture of the gap has been attempted by oxidation of graphene, in graphene oxides[2], by π-stacking of graphene layers, in few layers graphenes[3], or by their curvature, in curved graphenes[4]. Despite some reports on the use of some of these alternatives for light emission, luminescence and more particularly gain amplification has been proved to be still not sufficiently efficient for many purposes[5,6]. The bottom-up approach to graphene based on the preparation of the so-called nanographenes, graphene nanoribbons or molecule-based graphenes, has allowed a more rational design and exploitation of their semiconducting properties and correlatively of their luminescence features, although these have received lateral focus[7,8]. Among the several applications of luminescence, such as in sensing, imaging or electroluminescence, lasing has been less developed in organic materials, possibly because of their intrinsic fragility and rapid deterioration under intense optical pumping[9]. In the field of graphene inspired molecular materials, the possibility to obtain stimulated emission through the observation of amplified spontaneous emission (ASE) has been reported only very recently[10,11]. However, applications in real laser devices have not been realized yet.

The atomistic control and structural precision that can be achieved with orthodox organic synthesis of nanographenes opens the way for the preparation of nanographenes with tailored emission properties[7,8,12]. However, not all of these recipes are valid for laser applications, because more extended nanographenes generally show smaller band gaps beyond a limit that reduces the photoluminescence (PL) quantum efficiency. In this scenario, it seems that lasing[13], which requires strong luminescence, appears to be difficult to achieve on extended nanographenes. Furthermore, the possibility of having lasing action covering the whole or a large portion of the visible spectrum for versatile applications, is even more challenging and tough to obtain[14,15]. This is exemplified by the case of obtaining organic materials displaying ASE and lasing in the near-infrared (NIR) border of the visible region[16]. The difficulty relies on the fact that the small optical gap needed to promote NIR absorptions and emissions, simultaneously provokes fluorescence quenching[17]. This is also the case for extended π−conjugated nanographenes, so such accomplishment represents a challenging task.

In the field of organic thin-film lasers[9,18,19], those based on active waveguide films and distributed feedback (DFB) resonators, both made of solution-processed organic materials, are receiving major attention for their prospect as inexpensive, mechanically flexible, wavelength tunable, compact lasers with easy integration to other devices[20–26]. Within this context, a recent landmark was the demonstration of lasers based on a top-layer resonator (a resist layer with an engraved one-dimensional relief grating) fabricated by holographic lithography (HL), which in combination with highly efficient and photostable dyes, showed multi-color emission in a centimeter-size single device with excellent performance[25].

In the present work, we characterize the ASE properties of three recently synthesized nanographenes (denoted as FZn, with n = 1, 2, 3; chemical structures in Fig. 1a)[27], characterized for having their four edges with zigzag structures, and are able to develop lasers with them in the modality of top-layer resonator DFB lasers, being both, active film and resonator, prepared from solution. It is well-known that the formation and elongation of the zigzag edges (based on all-trans polyacetylene chains) of graphene nanoribbons is a very efficient way for maximizing red-shifts of the absorption/emission wavelength bands (i.e., ideal for

versatile color tunability)[28] in contrast to the limited π-electron delocalization in nanographenes with armchair edges (based on cis-trans polyacetylene substructures)[29]. Hence, the FZ1-3 derivatives of anthanthrene represent an ideal choice for color emission tunability and lasing, given that they are all constituted by four zigzag edges. This contrasts with a recent report of ASE[10] in a nanographene with the same number of benzenoid rings (i.e., 12) as FZ3, but constituted by a combination of armchair and zigzag edges, which results in a blue-shift of the relevant optical bands by 40–50 nm compared to ours. FZ3 would thus constitute the nanographene with the more red-shifted ASE (and DFB laser emission constructed based on it) found so far. It is well-known that in extended zigzag edged nanographenes, it is common to have mid-gap states with paramagnetic character (edged states), whose electronic characteristics are detrimental for light emission[30]. Noticeably, the size of FZ3 is optimal to provoke the maximal reduction of the band gap, but still insufficient to form edge states, which is beneficial for the permanence of luminescence and the properties thereof. The bulky aryl groups attached onto the zigzag edges in FZ1-3 not only stabilize the otherwise reactive species, but also suppress aggregation in solid state and enhance solubility in organic solvents.

Remarkably, the three ad-hoc designed nanographenes show efficient ASE emission from the blue to the NIR, and the three have been successfully implemented in DFB lasers. The fabrication of the DFB devices has followed a unique protocol developed recently in our laboratories[25]. These applications represent a significant finding in the field of molecular graphenes and broaden the versatility of these materials, not only as devices exploiting their semiconducting and magnetic properties, but also in systems based on light emission, such as lasers.

## Results

**Optical properties of nanographenes.** Thin films containing 1 wt % of FZn, dispersed in polystyrene (PS), used as a passive matrix, were spin-coated over transparent fused silica substrates. Film thickness, h, were adjusted to ensure that the waveguides support only fundamental transversal modes (electric and magnetic, $TE_0$ and $TM_0$, respectively), propagating with a high confinement factor ($\Gamma \approx 90\%$). This means choosing h values just below the cutoff thickness for the propagation of the first-order mode $TE_1$. Such election is convenient to minimize losses, and thus to optimize the ASE performance[31,32].

The absorption (ABS) and photoluminescence (PL) spectra (at room temperature, RT) of the prepared films are shown in Fig. 1b (all relevant optical parameters are listed in Table 1). PL quantum yield (PLQY) values as large as 82% have been measured, indicating the excellent potential of the prepared materials for solid-state emission (see Table 1). The ABS and PL spectra are mirror-like and show small stokes shifts from their strongest peaks (Fig. 1b). Their ABS/PL spectra cover a large part of the visible region from 452/456 nm in FZ1 to 545/547 nm in FZ2 and 668/676 nm for FZ3. The second replica of the main PL spectral bands in Fig. 1b appear at 485 and 588 nm in FZ1 and FZ2, respectively, whereas in FZ3 this weaker band is detected as a higher wavelength shoulder at around 740 nm.

In order to get insights on the origin of the PL transitions, we performed PL measurements at low temperature (80 K) in methyl tetrahydrofuran (THF) solution (see Fig. 1c), as well as Raman characterization, i.e., with resonant and pre-resonant conditions in solid state (Supplementary Fig. 1 and related discussion). It is seen that each vibronic component of the RT PL emissions resolve in subpeaks delineating two main vibronic progressions spaced by $\approx 1300\,cm^{-1}$ ($1324\,cm^{-1}$ in FZ1, $1326\,cm^{-1}$ in FZ2 and $1329\,cm^{-1}$ in FZ3) and $\approx 300\,cm^{-1}$ ($333\,cm^{-1}$ in FZ1,

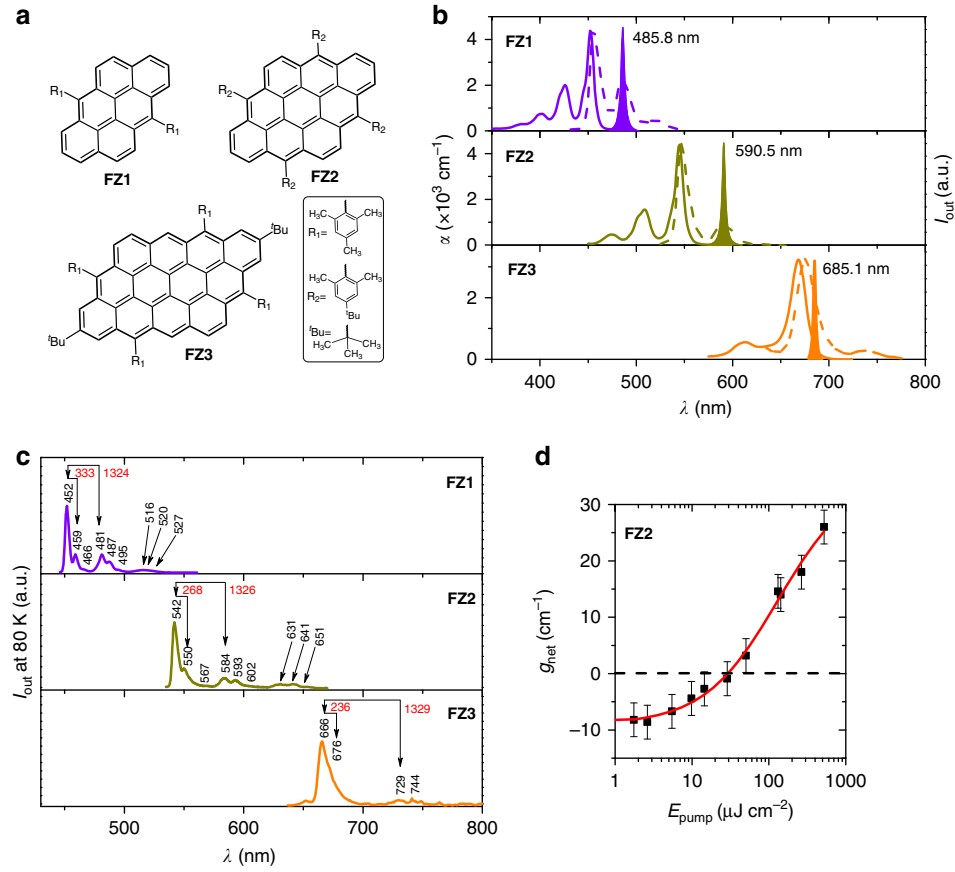

**Fig. 1** Chemical structures of nanographenes and their optical properties dispersed in polystyrene films. **a** Chemical structures of nanographenes FZ1, FZ2, and FZ3, respectively. **b** Optical properties at room temperature of PS films doped with 1 wt% of FZ1, FZ2, and FZ3, from top to down. Absorption coefficient, $\alpha$ (solid line, left axis), photoluminescence intensity (dashed line, right axis), and amplified spontaneous emission, ASE, intensity (filled area, right axis), versus wavelength, $\lambda$. **c** Low temperature (80 K) PL spectra of the three nanographenes in methyl tetrahydrofurane (methyl THF) solution. The wavelengths of each peak, and the spacing (in red, in $cm^{-1}$) for each vibrational progression are indicated. **d** Net gain coefficients, $g_{net}$, obtained from plots such as those of Supplementary Fig. 3, versus the pump energy density, $E_{pump}$, for a 1 wt% FZ2-doped PS film. The full line is a guide to the eye and its intersection with the $y$-axis corresponds to the loss coefficient ($k = 9 \pm 1\,cm^{-1}$). Errors in $g_{net}$ were estimated statistically as the standard deviation from measurements on several nominally identical samples. Source data are provided as a Source Data file in the Institutional Repository of the University of Alicante [http://hdl.handle.net/10045/92007]

### Table 1 Optical properties of FZ nanographene compounds dispersed in PS films (no resonator)

| FZ compound[a] | PLQY[b] [%] | $\lambda_{ABS\text{-}max}$[c] (nm) | $\lambda_{PL\text{-}max}$[d] (nm) | $h$[e] (μm) | $\lambda_{pump}$[f] (nm) | $\alpha\,[\lambda_{pump}]$[g] ($\times 10^3\,cm^{-1}$) | $t_p[\lambda_{pump}]$[h] (ns) | $\lambda_{ASE}$[i] (nm) | $FWHM_{ASE}$[j] (nm) | $E_{th}^{ASE}$[k] (μJ cm$^{-2}$) |
|---|---|---|---|---|---|---|---|---|---|---|
| 1 (1 wt%) | 82 | 402, 426, <u>452</u> | <u>456</u>, 485, 519 | 0.40 | 452 | 4.40 | 3.9 | 485.8 | 4 | 45 |
|  |  |  |  | 0.40 | 426 | 1.95 | 3.8 | 485.8 | 4 | 60 |
| 2 (1 wt%) | 73 | <u>474</u>, 509, 545 | <u>547</u>, 588 | 0.48 | 545 | 4.40 | 4.4 | 590.5 | 4 | 30 |
|  |  |  |  | 0.48 | 509 | 1.55 | 4.2 | 590.5 | 4 | 70 |
| 2 (3 wt%) | 31 | <u>474</u>, 509, 545 | <u>548</u>, 589 | 0.50 | 545 | 14.3 | 4.4 | 591.7 | 5 | 200 |
| 3 (1 wt%) | – | 613, <u>668</u> | <u>676</u>, 740 | 0.60 | 668 | 3.30 | 4.6 | 685.1 | 3 | ~2 × 10⁴ |
|  |  |  |  | 0.60 | 613 | 0.56 | 4.5 | 685.1 | 3 | ~3 × 10⁴ |

[a]Number of nanographene compound, among 1, 2 and 3; doping rate (error ~0.1 wt%) in PS indicated in brackets
[b]Photoluminescence quantum yield (error ~10%)
[c]Peak absorption wavelengths (maximum absorption peak is underlined)
[d]Peak photoluminescence wavelengths (maximum photoluminescence peak is underlined)
[e]Film thickness (error ~2%)
[f]Pump wavelength
[g]Absorption coefficient at $\lambda_{pump}$ (error ~2%)
[h]Pump pulse width at $\lambda_{pump}$
[i]Amplified spontaneous emission (ASE) wavelength (error is ± 0.5 nm)
[j]ASE linewidth (error is ±1 nm), defined as the full width at half maximum, FWHM, well above the threshold
[k]ASE threshold (error ~20%), determined from plots such as those of Supplementary Fig. 2, as the incident pump energy density at which the FWHM decays to half of its maximum value

268 cm$^{-1}$ in FZ2 and 236 cm$^{-1}$ in FZ3), see Fig. 1c (a similarly spaced third progression can be mentioned considering the very weak components at 466/495/527 in FZ1, at 567/602/651 nm in FZ2 and undetected in FZ3). These vibronic progressions both correspond approximately to main Raman peaks at 1321 cm$^{-1}$ in FZ1, 1279 cm$^{-1}$ in FZ2 and 1298 cm$^{-1}$ in FZ3 and at 342 cm$^{-1}$ in FZ1, 301 cm$^{-1}$ in FZ2 and 243 cm$^{-1}$ in FZ3 (Supplementary Fig. 1). The two series of the pre-resonant Raman bands correspond to in-plane CC stretching modes and to soft skeletal deformation vibrations, respectively, both mimicking the pattern of the structural/electronic changes from the ground electronic state to the emissive excited state.

In order to exploit the emission properties of FZ1-3 for laser applications, we have first evaluated the film ASE properties. The ASE phenomenon[33,34] is required to achieve lasing in waveguide-based devices, such as DFB lasers, and its observation is a signature of the existence of gain. It reflects on a narrowing of the PL spectrum at a given pump intensity, accompanied by a sudden increase of the emission intensity (see illustrative plots in Supplementary Fig. 2). For all the nanographenes, the ASE light collected from one edge of the films did not show preferential polarization in a given direction. The fact that the films support two waveguide modes, TE$_0$ and TM$_0$, whose polarizations are parallel and perpendicular, respectively, to the film edge from which light is collected, is an indication that light traveling in both modes, experiences gain due to ASE.

The ASE spectra of the nanographene-doped PS films are shown in Fig. 1b. The ASE linewidth, defined as the full width at half maximum (FWHM) is typically of a few nm. ASE peaks ($\lambda_{ASE}$) occur at 485.8, 590.5, and 685.1 nm, for FZ1, FZ2, and FZ3, respectively. They occupy a key position in the range of the three magic colors for lighting applications, blue color (450–490 nm) in FZ1 ($\lambda_{ASE} = 485.8$ nm), near the green color (520–560 nm) in FZ2 ($\lambda_{ASE} = 590.5$ nm) and red color (635–700 nm), at the edge of the NIR region, in FZ3 ($\lambda_{ASE} = 685.1$ nm). This is significant given the chemical compatibility among very similar parent molecules, which might allow the hypothetical preparation of composite materials with the three nanographenes in the same gain medium. It is worth comparing these results to those recently reported for a series of six carbon-bridged oligo(p-phenylenevinylene) (COPVn, with n = 1 to 6), dispersed in PS, which demonstrated an excellent ASE and DFB laser performance[35]. The COPV1-6 spectral emission range covers from 385 nm in COPV1 to 583 nm in COPV6, a wavelength segment of 150 nm for which it is needed six compounds. In the case of the FZ1-3 compounds, a more red-shifted part of the spectrum (from 452 to 668 nm) is covered, with a total wavelength segment of 228 nm, which is achieved with only three compounds. Note that within each of the COPVn (with n = 1 to 6) and FZn (with n = 1 to 3) series, when moving from one compound (with a given n), to the next (with n + 1), the same increment of eight π-electrons occurs with a largely distinctive impact in the wavelength modification (larger for the FZn compounds). This is a consequence of the different π-electron conjugation motif, one-dimensional (1D) in COPVn versus 2D (zigzag plus armchair π-delocalization) in FZn. The result for FZ3 is significant as it overcomes the inherent difficulty of small optical gap molecules as for emission efficiency due to the emergence and multiplication of non-radiative mechanisms of deactivation when entering in the NIR region of the electromagnetic spectrum.

Focusing now on the physical mechanisms underlying the appearance of ASE and according to the low temperature PL and pre-resonant Raman results (Fig. 1c and Supplementary Fig. 1, respectively), it is seen that the wavelengths at which ASE appears for the three compounds correspond approximately to well-defined peaks in the Raman spectra. Noticeably, in FZ1 and FZ2,

ASE seems to be activated by the more energetic Raman peaks (main vibronic progression) at around 1300 cm$^{-1}$, whereas in FZ3, it is the second satellite progression and the far-infrared Raman peak at 300 cm$^{-1}$ that appears to be responsible of the ASE activity. The approximate correspondence between the active vibronic progression and the vibrational bands in the pre-resonant (resonant) Raman spectra (see discussion in the supporting information file related to Supplementary Fig. 1) is indicative that the ASE is controlled by in-plane vibrational modes (i.e., with the same polarization of the active electronic transition) and that the electron-vibration coupling with the relevant excited state (from which ASE is produced) is controlled by high-energetic CC stretching modes in FZ1 and FZ2 and evolves towards a control by low-frequency molecular vibrations in FZ3. It is common to find the ASE peaks in organic dyes in the second vibronic replica of the emission spectra due to the strong reabsorption in the first one. Consequently, given the similarity of the emission spectra of the three samples in Fig. 1b, the appearance of ASE in the first vibronic replica in FZ3 might attend to a fundamental change in the electron-vibration mechanism along the relevant electronic excitation related to the fact that the electronic structure changes from small benzenoid aromatics (such as FZ1 and FZ2) to larger zigzag edged polycyclics (such as FZ3) where this starts to be affected by the activity of the edges states (the molecule becomes 2D π-electron delocalized). This prominent change in the underlined ASE/vibronic-Raman mechanism would affect the features of the emerging ASE properties, such as revealed in the ASE stability of FZ3 compared to FZ1/FZ2.

The ASE performance in terms of threshold (i.e., the incident pump energy/intensity at which ASE appears) is very good for FZ1 and FZ2 (see Table 1). ASE threshold values, expressed as incident energy density $E_{th}^{ASE}$ (or power density, $I_{th}^{ASE}$), determined from plots such as the one shown in Supplementary Fig. 2, are as low as 45 μJ cm$^{-2}$ (12 kW cm$^{-2}$) and 30 μJ cm$^{-2}$ (7 kW cm$^{-2}$), for FZ1 and FZ2, respectively. These values are lower than those recently reported for other nanographenes (60 μJ cm$^{-2}$)[10] and comparable to the ones of highly efficient laser dyes dispersed in PS, such as perylenediimides (PDI)[36] or COPVn[35]. A higher ASE threshold is obtained for FZ3 (Table 1), mainly attributed to the proximity between the absorption and the ASE emission that results in a much larger reabsorption.

For the best performing compound, FZ2, we have characterized the gain and losses, which are important parameters to provide a full description of the ASE performance of a given material. The net gain coefficients ($g_{net}$), determined from plots such as those shown in Supplementary Fig. 3 (see Methods for details) at different pump energy densities, $E_{pump}$, are plotted in Fig. 1d. At $E_{pump} = 52$ μJ cm$^{-2}$ (i.e., 11.8 kW cm$^{-2}$) a net gain $g_{net} = 3.3$ cm$^{-1}$ is obtained. Despite this value is slightly inferior than those of state-of-the-art dyes for lasing (whose PLQY ~100%), it can be considered reasonably good, considering the low dye doping rate used here (1 wt%) and also the lower PLQY values of these nanographenes (73% for the film based on FZ2). In any case, the $g_{net}$ value obtained for the FZ2 film is comparable to that of other PDI derivatives[37] successfully used in various types of DFB lasers (i.e., with gratings engraved by nanoimprint lithography on inorganic substrates[38]), or directly on the active film[39]; or with gratings engraved by HL over resist substrates[40]. The total loss ($k$), which is independent of $E_{pump}$, was determined from Fig. 1d by extrapolating the $g_{net}$ to $I_{pump} = 0$. The obtained value ($k = 9$ cm$^{-1}$) is similar to that of other laser dyes dispersed at a similar concentration in PS[36]. This indicates that at this low loading, the propagation properties are dictated by the polymer host.

The possibility to reduce the ASE threshold by increasing the nanographene doping rate in the matrix was explored by

preparing films with 3 wt% of FZ2 (see results in Table 1). Unfortunately, the PLQY decreases and the threshold increases, which is attributed to PL quenching due to dye aggregation and/or interaction. In view of these results, DFB devices will be based on 1 wt% doped films.

Remarkably, the nanographene films are very stable against air and moisture. Upon storage of the prepared films in air for several months, their ASE performance is preserved. Their ASE stability against photodegradation is also very large. ASE operational lifetimes of ~$10^5$ pump pulses under optical excitation (in ambient conditions and without encapsulation) in the same spot of the sample, at pump intensities several times above threshold, were measured even for FZ3, whose threshold is rather large. The high stability of these nanographenes arises from several factors: (i) Their own chemical structures, in which the most chemically reactive sites in the molecular peripheries are blocked with chemically inert and bulky aryl groups (kinetic blocking), as well as their inherent rigid molecular backbones, which restrict excited state conformational mobility and non-radiative processes. And (ii) Their use dispersed in a solid matrix, which prevent intermolecular interactions and reactions, thus minimizing PL quenching and degradation. Detailed quantitative photodegradation studies have been performed with the constructed DFB lasers and are discussed in the following section.

**Nanographene DFB Lasers.** The potential of the nanographenes under study for laser applications is demonstrated by the fabrication of a series of devices, which show emission across a wide range of wavelengths, at a low threshold and showing very long operational lifetimes. The device structure (see Scheme in Fig. 2a), includes a top-layer polymeric resonator, consisting of a water-soluble dichromated gelatin (DCG) photoresist layer with a 1D relief grating (engraved by HL), deposited on top of an active film of nanographene dispersed in PS. A distinctive feature of this device architecture is that $h$ is constant across the device because the diffractive grating that provides the feedback required for laser emission is in a separated layer. The DCG layer is processed from water, so the properties of the active layer keep unaltered by the resonator preparation on top of it. Additionally, this system enables multi-color emission within a single chip while keeping a low threshold[25], in contrast to other reported strategies[41–43] for wavelength tuning, which are either more sophisticated and/or impose limitation on other parameters, such as the threshold, which is highly dependent on $h$. The good laser tuning characteristics of the devices prepared here are a consequence of their large size (cm), the versatility offered by HL and the fact that $h$ is constant across the device.

All the DFB devices prepared here are 1D and operate in the second-order of diffraction, that is $m = 2$ in the Bragg condition (Eq. 1)[9,18,19]:

$$m \, \lambda_{Bragg} = 2 \, n_{eff} \, \Lambda \qquad (1)$$

where $n_{eff}$ is the effective refractive index of the waveguide (which depends on $h$ and on the refractive indexes of film, substrate, and cover) and $\Lambda$ is the grating period. Second-order DFBs are very attractive for certain applications, such as sensing[44–47], because light is coupled out of the film mainly in a direction perpendicular to the film by first-order diffraction, at a wavelength $\lambda_{DFB}$ close to $\lambda_{Bragg}$. Considering light traveling in a given waveguide mode, coupled mode theory[48] predicts that for pure index gratings (this is the case for the lasers prepared here, because $h$ is uniform across the device and the grating is in a separated layer), the wavelength that exactly satisfies Eq. (1) cannot propagate in the film. So, a photonic stop-band centered at $\lambda_{Bragg}$ appears, and lasing oscillates on a pair of wavelengths,

one at either edge of the dip. However, in the case of second-order devices, the peak with the lower wavelength has a larger threshold due to radiation losses[49]. Accordingly, single-mode emission at the peak of the longer wavelength is observed.

The geometrical and performance parameters of the prepared lasers are collected in Table 2. The grating period $\Lambda$ was varied in the range 300–450 nm to obtain lasing at wavelengths close to the maximum gain for each compound (that at which ASE emission occurs, $\lambda_{ASE}$; Table 1). The lasers emit at different regions of the visible and in the border of the NIR (see laser spectra in Fig. 2b and Table 2). Particularly, $\lambda_{DFB}$ values are in the ranges 482–495 nm (FZ1), 584–601 nm (FZ2) and 674–689 nm (FZ3). For a given compound, $\lambda_{DFB}$ was tuned by changing $\Lambda$. Noticeably, we have a very fine design control over the desired $\lambda_{DFB}$ value thanks to the versatility of the HL technique used to engrave the gratings; and also because $h$ is uniform across the device. For FZ2, we prepared a larger set of devices aiming to determine the tunability range, which results to be of around 17 nm, i.e., $\lambda_{DFB}$ covers from 584.1 nm (device $2^A$) to 601.3 nm (device $2^G$). Importantly, the threshold keeps at a low value for most of this range (see Table 2). Note that the tunability range obtained for the FZ2 lasers could be further enlarged, but this would imply an increase of threshold as gain decreases when $\lambda_{DFB}$ deviates from $\lambda_{ASE}$. In addition, the lasers emitting at lower wavelengths face limitations that arise from self-absorption. As it is seen in Fig. 2b and Table 2, the lasers show either one or two peaks. Whatever case, they are very narrow (< 0.13 nm, limited by spectral resolution), as illustrated in Fig. 2c. These peaks are laser modes, which can be easily associated with a given mode of the waveguide ($TE_0$ or $TM_0$), simply by looking at their polarization properties. A peak whose light is polarized parallel to the grating lines corresponds to $TE_0$, while $TM_0$ is associated with peaks with light polarized perpendicularly to the grating lines, and this polarization is independent on the type of polarization of the pump beam. In addition, this can be predicted by calculating the effective index of the mode and then through Eq.1 to obtain $\lambda_{Bragg}$[35,47]. As it is seen in Fig. 2b and Table 2, for the lasers showing two peaks, the corresponding $\lambda_{DFB}$ values are both close to $\lambda_{ASE}$ and their thresholds are similar. On the other hand, as $\lambda_{DFB}$ deviates from $\lambda_{ASE}$, at some point the laser oscillates only in one mode, particularly in the one that is closer to $\lambda_{ASE}$ (i.e., associated with $TE_0$ if $\lambda_{DFB} < \lambda_{ASE}$, or $TM_0$ if $\lambda_{DFB} > \lambda_{ASE}$), because its threshold is significantly lower than that of the other one. Interestingly, in previous studies of organic DFB lasers with gratings engraved in the substrate, emission associated with TM waveguide modes was not observed, because their thresholds were generally larger than those of the TE modes (they could eventually be seen, but under a very strong excitation)[22,35,38,40,47,50]. This might be a consequence of the different architecture used here. Further studies to clarify this are currently under way.

Figure 2d shows images of the blue and yellow light emitted by two of the prepared lasers ($1^B$ and $2^C$), based on FZ1 and FZ2, respectively. No image is shown for a laser based on FZ3 because the laser intensity is not large enough (relative to noise, which is rather high due to the strong pump intensity needed to operate), to obtain it with reasonable quality. The beam divergence observed in the direction perpendicular to the grating lines is ~ $5 \cdot 10^{-3}$ rad.

In terms of laser threshold, the performances of FZ1 and FZ2 are excellent, and more particularly that of the latter (see Table 2). Threshold data are extracted from plots of the emission linewidth, such as those shown in Fig. 3a (the corresponding output intensity versus $E_{pump}$ plots are shown in Fig. 3b). Low values between 11 and 18 $\mu$J cm$^{-2}$ (i.e., 3–4 kW cm$^{-2}$) have been obtained for the various devices based on FZ2 ($2^C$ to $2^F$), whose $\lambda_{DFB}$ emissions are in the approximate range 590–600 nm. These values are comparable to state-of-the-art DFB lasers based on

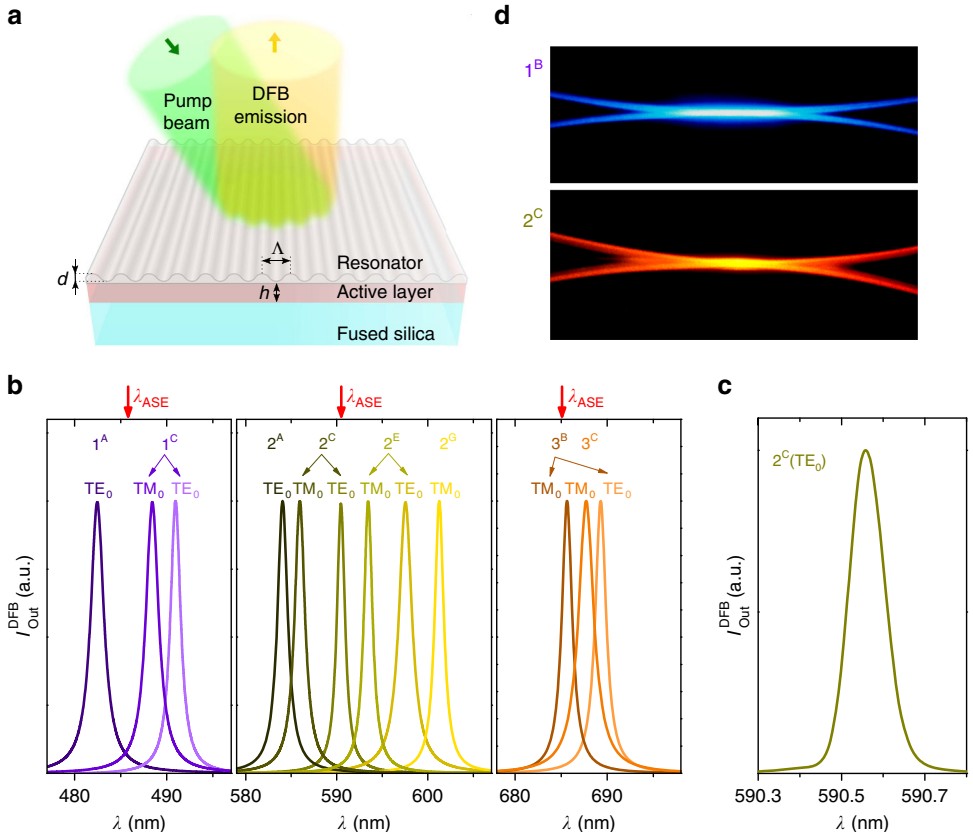

**Fig. 2** Distributed feedback (DFB) laser architecture and spectral properties of nanographene lasers. **a** Sketch of the DFB device, consisting of a top-layer polymeric resonator with an engraved relief grating ($\Lambda$, grating period; $d$, grating depth), located over an active film ($h$, film thickness) of nanographene dispersed in polystyrene (PS), deposited on a fused silica substrate. The excitation and collection geometries are shown by arrows. **b** Spectra of DFB lasers based on nanographene-doped PS films. The number on the device label refers to the nanographene molecule used among FZ1, FZ2, or FZ3; the letters on the labels refer to devices with different geometrical parameters (listed in Table 2). The emitted laser light consists of either one or two peaks (laser modes), each associated with a given waveguide mode of the film, $TE_0$ or $TM_0$, whose light is polarized parallel or perpendicular to the DFB grating lines, respectively. Top red arrows indicate the ASE wavelength for each case (exact values in Fig. 1b). **c** High-resolution spectrum of one of the laser peaks on an expanded scale (device 2$^C$, peak associated with the $TE_0$ waveguide mode). **d** Images of the total light emitted by devices 1$^B$ and 2$^C$. Source data are provided as a Source Data file in the Institutional Repository of the University of Alicante [http://hdl.handle.net/10045/92007]

PDI[36] and COPV$n$[35]. For FZ3, thresholds are larger, in accordance with the ASE thresholds discussed previously. The laser slope efficiency (LSE) of devices based on FZ1 and FZ2 were also obtained (Table 2 and Supplementary Fig. 4). The rather low values are typical of active materials based on dye-doped polymers at low loadings such as the ones used here (1 wt%). Noticeably, thanks to the top-layer resonator configuration used, the obtained LSE are expected to be larger than the ones achievable with other resonator configurations[25].

The mechanisms involved in the lasing process are analyzed through the evolution of the emission spectrum as $E_{pump}$ is increased (see Fig. 3c). At a low $E_{pump}$ (below the threshold) the characteristic Bragg dip is observed (this corresponds to $\lambda_{Bragg}$ in Eq. 1). As $E_{pump}$ increases, a narrow lasing peak raises at a wavelength a few nanometer above this value. This is in accordance with predictions of coupled mode theory for devices with pure index coupling, such as the top-layer DFB lasers used in this work[48–50]. Nevertheless, the observation of band-edge lasing is common in second-order 1D devices with other DFB architectures, even with modulated film thickness. In those cases, lasing is dominated by index coupling, although gain coupling is also present[9,18,19,50].

A remarkable property of the prepared nanographene lasers is their outstanding operational lifetimes. As seen in Fig. 3d, the laser intensity of devices based on FZ2 and FZ3 keep practically

unaltered after $2 \times 10^5$ pump pulses at moderate conditions (MP, accounting for moderate pump) relative to their corresponding thresholds ($E_{pump} \approx 4 \times E_{th\text{-}DFB}$). This result is very significant in comparison to state-of -the-art highly photostable dyes, such as COPV$n$[35] or PDI[36,38], which show a similar performance under excitation at somewhat softer conditions (i.e., at two times above threshold). To better quantify this property, as well as to enable proper comparisons among different materials, excitation was also done at more extreme pump (EP) conditions and with the same pump energy for all of the devices ($E_{pump} = 2 \times 10^4\ \mu J\ cm^{-2}$) and determined the corresponding DFB half-lifes, $\tau_{1/2}^{DFB}$ (see Fig. 3d and Table 2). As expected, under such intense excitation, the operational lifetimes of FZ1 and FZ2 are reduced, although the value for FZ2 is still quite good ($7.5 \times 10^3$ pump pulses). However, the most significant result is the enormous lifetime of FZ3, for which the MP and EP conditions are in fact equivalent due to its large threshold. As already mentioned, the laser intensity keeps approximately the same after such long time under excitation. The large DFB operational lifetime is a consequence of the high stability of the nanographene films, as discussed in the above ASE section. The outstanding stability of FZ3, whose threshold is larger, might be related to the different mechanism (compared to FZ1 and FZ2) involved in the observation of ASE, as suggested by the Raman data. This might eventually have implications in the device stability.

**Table 2 Parameters of top-layer resonator distributed feedback (DFB) lasers based on nanographenes as active media**

| Laser device[a] | $h$[b] (μm) | $\Lambda$[c] (nm) | $\lambda_{pump}$[d] (nm) | $\lambda_{DFB}$[e] (nm) | $E_{th}^{DFB}$[f] (μJ cm$^{-2}$) | $I_{th}^{DFB}$[f] (kW cm$^{-2}$) | $\tau_{1/2}^{DFB}$[g] (MP) (pump pulses) | $\tau_{1/2}^{DFB}$[h] (EP) (pump pulses) | LSE[i] (%) |
|---|---|---|---|---|---|---|---|---|---|
| 1$^A$ | 0.40 | 307 | 452 | 482.5 (TE$_0$) | 90 | 23 | | | |
| 1$^B$ | 0.40 | 311 | 452 | 484.4 (TM$_0$) | 50 | 13 | | | |
| | | | | 487.1 (TE$_0$) | 50 | 13 | $2 \times 10^4$ | $7.20 \times 10^2$ | 0.77 |
| 1$^C$ | 0.40 | 314 | 452 | 488.4 (TM$_0$) | 60 | 15 | | | |
| | | | | 491.0 (TE$_0$) | 95 | 24 | | | |
| 1$^D$ | 0.40 | 316 | 452 | 491.3 (TM$_0$) | 100 | 25 | | | |
| | | | | 494.9 (TE$_0$) | 160 | 40 | | | |
| 2$^A$ | 0.48 | 378 | 545 | 584.1 (TE$_0$) | 120 | 27 | | | |
| 2$^B$ | 0.48 | 380 | 545 | 588.4 (TE$_0$) | 25 | 6 | | | |
| 2$^C$ | 0.48 | 382 | 545 | 586.0 (TM$_0$) | 25 | 6 | | | |
| | | | | 590.5 (TE$_0$) | 11 | 3 | $> 2.5 \times 10^5$ | $7.5 \times 10^3$ | 0.62 |
| 2$^D$ | 0.48 | 383 | 545 | 588.1 (TM$_0$) | 30 | 7 | | | |
| | | | | 592.8 (TE$_0$) | 18 | 4 | | | |
| 2$^E$ | 0.48 | 386 | 545 | 593.4 (TM$_0$) | 16 | 4 | | | |
| | | | | 597.5 (TE$_0$) | 15 | 3 | | | |
| 2$^F$ | 0.48 | 388 | 545 | 595.1 (TM$_0$) | 16 | 4 | | | |
| | | | | 599.7 (TE$_0$) | 15 | 3 | | | |
| 2$^G$ | 0.48 | 393 | 545 | 601.3 (TM$_0$) | 75 | 17 | | | |
| 3$^A$ | 0.60 | 437 | 610 | 674.5 (TE$_0$) | $\sim 6 \times 10^3$ | $\sim 1.5 \times 10^3$ | $> 1.5 \times 10^5$ | | |
| 3$^B$ | 0.60 | 446 | 668 | 685.7 (TM$_0$) | $\sim 3 \times 10^3$ | $\sim 1 \times 10^3$ | | | |
| | | | 610 | 689.3 (TE$_0$) | $\sim 9 \times 10^3$ | $\sim 2 \times 10^3$ | | | |
| | | | 668 | 689.3 (TE$_0$) | $\sim 4 \times 10^3$ | $\sim 1 \times 10^3$ | $> 2 \times 10^5$ | $> 2 \times 10^5$ | |
| 3$^C$ | 0.60 | 449 | 668 | 687.5 (TM$_0$) | $\sim 6 \times 10^3$ | $\sim 1.5 \times 10^3$ | | | |

[a]The DFB device consists of an active film of polystyrene doped with 1 wt% (error ~0.1%) of nanographene with a top-layer of dichromated gelatine with an engraved relief grating. The number in the device label refers to the nanographene compound (1, 2, and 3, for FZ1, FZ2, and FZ3, respectively). The letters (A, B, C, and D) refer to devices with different grating periods, thus emitting at different wavelengths
[b]Film thickness (error ~2%)
[c]Grating period (error ~0.5%)
[d]Pump wavelength
[e]DFB wavelength (error is ± 0.5 nm) for each laser peak (laser mode) observed in the emission spectrum. The waveguide mode (TE$_0$ or TM$_0$) to which the laser mode is associated is shown in brackets. The emitted laser light is polarized parallel (for TE$_0$) or perpendicular (for TM$_0$) to the DFB grating lines
[f]DFB threshold (error ~10%), determined from Fig. 3a, expressed as energy density, $E_{th\text{-}DFB}$, or power density, $I_{th\text{-}DFB} = E_{th\text{-}DFB}/t_p$ ($t_p$ is the pump pulse width, values in Table 1)
[g]DFB operational lifetime, characterized by the photostability half-life $\tau_{1/2}^{DFB}$ (determined from Fig. 3d) measured in air under a moderate pump (MP) of $E_{pump} \sim (4 \times E_{th\text{-}DFB})$ at 10 Hz (error ~10%)
[h]Same as in g, but under an extreme pump (EP) of $E_{pump} \sim 2 \times 10^4$ μJ cm$^{-2}$, at 10 Hz (error ~10%)
[i]Experimental laser slope efficiency, obtained from the total emitted light (at any polarization), determined from Supplementary Fig. 4 (error ~5%)

## Discussion

The ASE properties of three nanographenes (FZ$n$, with $n = 1, 2, 3$) derived from anthanthrene having four zigzag perimetral edges have been studied and interpreted. Results reveal a change in the ASE mechanism as a function of the molecular size, which is a key aspect to consider for further exploitation of these materials for lasing applications. They show outstanding stability, both chemical and against photodegradation, which is a consequence of effective kinetic blocking of the most reactive zigzag edges by bulky aryl groups, as well as the rigid, largely delocalized backbone. Also, the high dilution (at 1 wt%) of the compounds in a polymer matrix is key to avoid molecular interactions and reactions, which are known to be detrimental for the PL efficiency and stability of molecular compounds[36]. Designing synthetic strategies to functionalize nanographenes, as to avoid those processes, would be of great interest. For such purpose, the existing experience in the field of organic lasers might be useful[9,35,36,51,52]. Photophysical studies by means of transient absorption and pump-probe experiments[10,11], would aid to get insights on the gain and photodegradation mechanisms in these compounds.

The studied nanographenes show ASE at wavelengths within the visible spectral region and approaching the NIR. Particularly at around 486, 590, and 685 nm, for FZ1, FZ2, and FZ3, respectively. By using small-size highly emissive polycyclic aromatics such as anthracene and pyrene derivatives, ASE in the vicinity of the ultraviolet (UV)–Visible (Vis) region can be achieved[53]; whereas by using more extended FZ$n$ analogs, ASE takes place in the opposite border of the VIS region (Vis–NIR).

For NIR lasing, it is important to develop new zigzag edged nanographenes showing at least moderate PLQY in solid state. Synthetic efforts along this line are ongoing in the laboratories of some authors of the present work.

The versatility of nanographenes for lasing is demonstrated by the preparation of solution-processed DFB devices. Their performance is outstanding in terms of operational lifetime ($> 2 \times 10^5$ pump pulses for FZ2 and FZ3). Their thresholds are quite good, considering the polymeric nature of the resonator, i.e., down to 11 μJ cm$^{-2}$ (3 kW cm$^{-2}$), for devices based on FZ2. An important factor concerns the uniqueness of the top-layer polymeric resonator architecture used in these DFB devices, which provides advantages with respect to other DFB modalities.

Considering that the nanographene field is very recent, the capabilities of nanographenes for all-solution-processed lasers demonstrated in this work, open new avenues for further developments and studies spanning across various scientific disciplines: (i) From the point of view of the chemical synthesis of nanographenes[54], several aspects of interest are: the development of stable nanographenes with high PLQY and functionalized with proper peripheral substitutents to achieve good solubility and to prevent aggregation and/or molecular interactions, as to allow larger loading rates in the matrix[35,51,52]; the design of nanographenes emitting at wavelengths in the UV and NIR regions of the optical spectrum, the latter being of particular importance for other related areas, such as that of plasmonic lasers[55,56]; the preparation of nanographenes showing good properties simultaneously for light emission and charge transport, which would be of interest for the purpose of achieving diode (electrically

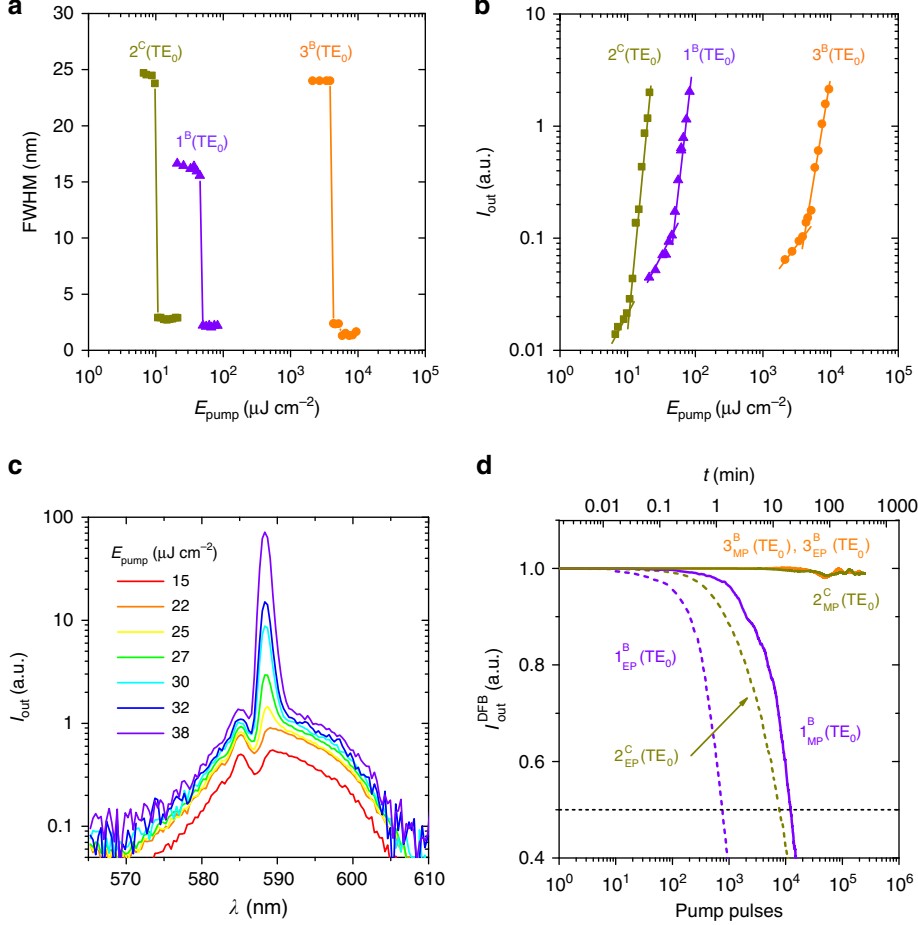

**Fig. 3** Threshold and operational lifetime performance of nanographene lasers. **a** Emission linewidth, defined as the full width at half maximum (FWHM), as a function of the pump energy density, $E_{pump}$, for the laser peaks associated with the $TE_0$ waveguide modes of devices $1^B$, $2^C$, and $3^B$. Full lines are guides to the eye. The DFB threshold for each case, $E_{th\text{-}DFB}$, is defined as the $E_{pump}$ at which the FWHM decays to half of its maximum value ($E_{th\text{-}DFB}$ values in Table 2). **b** Log–Log plots of output intensity, $I_{out}$, as a function of the pump energy density, $E_{pump}$, for the same laser peaks shown in panel **a**). For each case, $I_{out}$ is collected at the corresponding $\lambda_{DFB}$. Full lines are guides to the eye. **c** Evolution of the laser spectrum with $E_{pump}$ for device $2^B$. **d** DFB intensity, $I_{out}^{DFB}$, versus time, $t$, and versus the number of pump pulses (top and bottom axes, respectively) for the same laser peaks shown in panel **a**) within this figure. The operational lifetime ($\tau_{1/2}^{DFB}$) for each case is defined as the time (or number of pump pulses), at which $I_{out}^{DFB}$ decays to half of its initial value. Excitation was done at the same spot of the device in air under a moderate pump (MP, full lines) of ($E_{pump}/E_{th\text{-}DFB}) \approx 4$, or under an extreme pump (EP, dashed lines) of $E_{pump} = 2 \times 10^4\,\mu J\,cm^{-2}$. Source data are provided as a Source Data file in the Institutional Repository of the University of Alicante [http://hdl.handle.net/10045/92007].

pumped) lasers[9,18,19]; (ii) With regards to device processing and nanofabrication, further studies with lasers based on the top-layer resonator architecture, aimed at implementing all-organic devices (including the substrate)[23,24] would be a step forward to real applications[26]; (iii) In the area of Physical Chemistry, performing photophysical experiments and quantum-chemical calculations would be useful to understand the mechanisms involved in the optoelectronic properties of these compounds. At this respect, contributions from the Condense Matter Physics community would also be very valuable, given its intense activity in the area of Graphene[1]; (iv) Finally, nanographenes with light emitting properties (including lasing) might provide an interesting bottom-up approach to the emerging field of 2D materials[57], to date mainly prepared by post-synthesis nanopatterning and nanofabrication techniques[58].

## Methods
**Synthesis**. Compounds FZ1, FZ2, and FZ3 were synthesized by multi-step Pd-catalyzed C–C coupling, Friedel-Crafts type alkylation and oxidative dehydrogenation. The detailed synthetic procedures and characterization data have been clearly described in our previous report[27].

**Low-temperature photoluminescence in solution**. Solutions of the compounds in 2-methyl tetrahydrofuran were prepared to obtain PL spectra at different temperatures from room conditions to 80 K in a cryostat OPTISTAT from Oxford instruments. All solvent used were of spectroscopic grade purchased from Aldrich. Emission and excitation spectra were measured using a spectrofluorometer from Edinburgh Analytical Instrument (FLS920P) equipped with a pulsed xenon flash-lamp, Xe900, of 400 mW.

**Raman measurements**. FT–Raman spectra for the various nanographenes in solid state (powder) were measured using the RAMII FT–Raman module of a VERTEX 70 FT-IR spectrometer. A continuous-wave Nd–YAG (YAG: yttrium aluminum garnet) laser working at 1064 nm was employed for excitation, at a laser power in the sample not exceeding 30 mW. Raman scattering radiation was collected in a back-scattering configuration with a standard spectral resolution of 4 cm⁻¹. Two thousand scans were averaged for each spectrum. The Raman spectra recorded using the 785 nm excitation were collected by using the $1 \times 1$ camera of a Bruker Senterra Raman microscope by averaging spectra during 50 min with a resolution of 3–5 cm⁻¹. A charged-coupled display camera operating at –50 °C was used.

**Thin-film and DFB resonator preparation**. Thin films of PS doped with FZ1, FZ2, or FZ3 (1 wt%) were prepared by spin coating, using toluene as solvent, over $2.5 \times 2.5\,cm^2$ commercial transparent $SiO_2$ substrates (i.e., quartz plates). Film thickness was determined from the interference pattern observed in the region of very low absorption of the absorption spectrum[59].

The fabrication of the DFB resonators was performed following several steps: (1) Photoresist layer deposition: a hot water solution (40 °C) with 2.2 wt% of DCG was spin-coated over the active film; (2) Grating recording: 1D gratings were recorded by HL with light from an Ar laser emitting at 364 nm, using a mirror attached with a 90° angle to the sample holder in a simple and stable set-up[25,60]; (3) Development: the DCG layers were desensitized in a cool water bath (15 °C) and the relief grating was formed by dry development in an oxygen plasma using the surface treatment machine Diener Zepto. The resultant relief grating has a depth of 90 nm.

**Optical experiments in nanographene films and devices**. The absorption and PL spectra of the active films were obtained in samples without DFB resonator by using a double-beam Jasco V-650 spectrophotometer and a Jasco FP-6500 spectrofluorimeter, respectively. Quantum yield PL measurements (PLQY) were performed with a Jasco ISF-834 integrating sphere attached to the fluorimeter.

ASE and DFB measurements were performed using as excitation source an optical parametric oscillator (OPO) pumped with a pulsed (10 Hz repetition rate) Nd:YAG laser operating at 355 nm. The pump wavelength ($\lambda_{pump}$) for each film was selected in order to match a peak of maximum film absorption ($\lambda_{pump}$ values, and their corresponding temporal pulse widths, $t_p$, are listed in Tables 1 and 2). The pump energy incident over the sample was varied using neutral density filters. For ASE characterization (spectra and threshold), films without resonator are excited at normal incidence with a beam shaped to a stripe of $3.5 \times 0.5$ mm$^2$ by means of a cylindrical lens; and the emitted light was collected from the edge of the film with an optical fiber coupled to an Ocean Optics USB2000 + UV–VIS fiber spectrometer with a spectral resolution of 1.3 nm. For FZ2, the net gain coefficients ($g_{net}$) at different pump energy densities ($E_{pump}$) are determined using the variable stripe length (VSL) method (Supplementary Fig. 3)[33]. We used the same set-up as for the ASE characterization, but using an adjustable slit to change the excitation stripe length, $l$, (from 0.01 cm to 0.3 cm). For a given $I_{pump}$ ($I_{pump} = E_{pump}/t_p$) the plot of the output intensity ($I_{out}$) versus $l$ was fitted to Eq. [2], using as fitting parameters $g_{net}$, and $A$ (parameter related to the cross section for spontaneous emission):[13,33]

$$I_{out}(\lambda) = \frac{A(\lambda)I_{pump}}{g_{net}(\lambda)}\left(e^{g_{net}(\lambda)l} - 1\right) \qquad (2)$$

Both, $A$ and $g_{net}$ depend on the wavelength ($\lambda$), and the latter is related to the gain, $g$, through $g_{net} = g - k$, where $k$ is a coefficient accounting for the total waveguide loss. The $k$ coefficient of the film, which is independent of $I_{pump}$, is obtained by representing $g_{net}$ as a function of $I_{pump}$ and extrapolating the $g_{net}$ value at $I_{pump} = 0$, at which $g_{net} = -k$[36,61]. For DFB characterization, the beam is elliptical (with a minor axis of 1.1 mm and area of 1.0 mm$^2$) and is incident at ~30° with respect to the normal to the device surface; and the emitted light is collected perpendicularly to the film surface with an optical fiber coupled to the spectrometer. For a detailed inspection of the spectral laser shape, a spectrometer with higher resolution (0.14 nm) was used (Ocean Optics MAYA2000). Laser slope efficiency (LSE) measurements were performed in the set-up used for laser characterization. Total output and pump pulse energies were measured with high sensitivity energy detectors Ophir PD10-C and PD10-pJ-C (resolutions of 1 and 0.01 nJ, respectively). The emitted laser light was collected by a 20 cm focal length lens and focused on the detector. LSE values were obtained by a linear fit of the output energy versus the pump energy curves, such as those shown in the Supplementary Fig. 4. The beam divergence was determined from direct measurements of the far-field pattern in a direction perpendicular to the resonator grating lines. For each of the compounds, we performed measurements on various nominally identical samples, aiming to ensure reproducibility of the ASE and DFB parameters.

**Reporting summary**. Further information on research design is available in the Nature Research Reporting Summary linked to this article.

## Data availability
The source data underlying Figs. 1b–d, 2b–c, 3a–d and Supplementary Figs. 1, 2, 3, and 4 are provided as Source Data files in the Institutional Repository of the University of Alicante [http://hdl.handle.net/10045/92007]. Other data supporting the findings of this manuscript are available from the corresponding authors upon reasonable request.

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

## Acknowledgements

The Alicante team acknowledges support from the Spanish Government (MINECO) and the European Community (FEDER) through grant no. MAT2015-66586-R. The researcher R.M.-M. has been partly supported by a MINECO FPI fellowship (no. BES-2016-077681). I. Garcés, V. Esteve, and J.M. Iglesias are also acknowledged for technical assistance. The work at the University of Málaga is supported by MINECO FEDER project reference CTQ2015-69391. J.W. acknowledges financial support from the MOE Tier 3 program (MOE2014-T3-1-004) and NRF Investigatorship Award (NRF-NRFI05-2019-0005).

## Author contributions

J.C., M.A.D.-G., and J.W. conceived the study, interpreted the data, and co-wrote the paper. V.B., J.M.V., and J.A.Q. fabricated the DFB resonators. V.B., R.M.-M., M.M.-V., and P.G.B. prepared the nanographene films and performed the optical, ASE, and DFB experiments. F.G.G. and J.C. carried out the low-T PL and Raman experiments. Y.G. and J.W. performed the synthesis of the compounds. All authors discussed the results and commented on the manuscript.

## Additional information

**Competing interests:** The authors declare no competing interests.

