## [Peer Review File · Nature Communications]

Reviewers' comments:

Reviewer #1 (Remarks to the Author):

Key results:

The manuscript by Bona et al. contains an interesting report on a multidisciplinary investigation of certain zig-zag type polycyclical aromatic molecules (nanographenes) which exhibit amplified spontaneous emission (ASE) and which can be used – when dispersed into polystyrene- as active media in optically pumped distributed feedback (DFB) lasers.

Albeit the different aspects of the manuscripts are not completely new, the manuscript is a very high quality piece of work bringing together a quite unique combination. It demonstrates for the first time the use of this class of materials in DFB-lasers. A particularly impressive result is the realization of lasers spanning a rather large range of wavelengths from 485 nm to 685 nm.

Validity:

The paper is well written and it contains very thorough and technically sound work. The group of authors brings together a long standing expertise ranging from the synthesis to device engineering and characterization. The last author has done pioneering work on different laser materials.

Originality and significance:

The paper does neither report on a completely new material nor a completely new photophysics but it brings together a very interesting materials class with a rather sophisticated and clever device structure and a very thorough investigation of the optical gain behavior. The results are very significant and will inspire many follow up activities in the field of nanographenes.

Methodology:

The data, the experimental procedures and the methodology to draw conclusions, the use of error bars, referencing of previous literature look very reasonable and convincing.

I have a couple of minor comments on smaller mistakes, typos or unconvincing parts of the manuscript which need/should be addressed prior to publications of this nice manuscript:

- in the affiliation part, the first “d” should be a “b”.

-On page 7, the authors are discussing the recent work by Morales-Vidales (actually with the same last author: “The emission spectrum covered by these COPVn ...” In here, it is somehow negatively discussed with a hint that six compounds were needed for a certain wavelength range while only three compounds are needed for the nanographenes? I do not understand this. If it is only about the overall range, it is always only a discussion of TWO compounds, namely the most red-shifted and the most blue-shifted one. If it is about spectral tunability, the paper needs more data. The tunability of the individual compounds (Fig. 2b) seems to be rather limited.

- on page 12, the authors claim that 700 nm are out of the “sensibility” (which I guess should read “sensitivity”) of the human eye. While the sensitivity goes down towards 780 nm, the general statement per se by the authors is wrong.

- The weakest part of the paper is the part “Discussion” which is more an extended conclusion as nothing really new is discussed but rather the statements from the results part of the paper are repeated. Besides the nice findings which are very interesting from an application oriented point of view, I personally would expect a little bit more of a discussion of the underlying photophysics or a discussion of future research directions. In any case, I am expecting a more thorough discussion of the rather surprising stability results. Is there any idea about the underlying mechanisms which lead to such differences in the lifetimes, especially the extended one for FZ3?

In summary, this is an important piece of science with high impact and I suggest to accept the paper after minor but mandatory revision.

Reviewer #2 (Remarks to the Author):

The authors report nanographene-based lasers from three zigzag edged polycyclic aromatics. The detailed experimental are presented with proper explanations for their mechanisms. The authors demonstrate that the nanographene can efficiently provide absorption/emission wavelength bands, covering a large part of the visible spectrum, and approaching very much to the near-infrared region. These results and fabrication method could promote the application of graphene in exploring excellent laser devices. In my opinion, this manuscript is suitable for publication in Nature Communications.

More comments and suggestions are listed below.

(1) In the manuscript, the absorption/emission wavelength bands of three nanographene molecules are studied. They show that the optical spectrum depends on the molecular size. In the three nanographene molecules, the FZ1 corresponds to the spectrum region with the shortest wavelength. Can nano-graphene further achieve shorter wavelengths? In addition, If we continue to get longer wavelengths by increasing the size of graphene, will we run into other constraints? What is the limitation of emission wavelength based on this mechanism? I suggest that the author discuss these issues in detail.

(2) In the manuscript, It is mentioned that the film thickness, h (values for all the films listed in Table 1) were adjusted to ensure that the waveguides support only one transversal electric mode (TE₀). The transverse magnetic mode (TM₀) might also be supported for the film thickness given in Table 1. The author should check this statement.

Reviewer #3 (Remarks to the Author):

The authors described the demonstration of solution-processed nanographene distributed feedback lasers with the narrow single mode emission in the broad spectral range by using passive polymer films and resonators. The obtained low threshold laser devices are flexible and low-cost, which is attractive for practical applications. Meanwhile, these devices behave the long lifetimes and low thresholds. Overall, this work is quite interesting and potentially important for the development of practical flexible lasers. Thus, I would like to consider its publication after addressing the following concerns.

1. The spectral narrowing data of lasers should be provided in the main text or Supplementary. Meanwhile, the cavity mode of the resonant cavity should be measured or further discussed. It will be better if the authors show the linewidth narrowing of the cavity mode above the threshold rather than the narrowing of the spontaneous emission band.
2. For the discussion on the transitions in Figure 1c, some of vibronic progressions do not agree well with the measured Raman shifts quantitatively. Thus, the authors may not directly correspond these vibronic progressions with the observed Raman features since not all vibrational modes can be reflected by Raman scattering data.

3. For Figure 3a, to visualize the threshold kink, it will be better to show the typical Log-Log plot instead of the used Log-Linear plot. The measured output intensity is collected from the full emission range or the ASE spectral range?

4. Does the polarization of the laser device vary with that of the pump laser beam or only along the direction of the grating lines no matter which incident polarization of the pump beam is used? Such information is helpful to determine the origin of the lasing polarization.

5. The amplified spontaneous emission of the nanographene doped polymer film is linearly polarized, partially polarized or unpolarized?

Reviewers' comments and responses:

Reviewer #1 (Remarks to the Author):

This paper reports the observation of amplified spontaneous emission and distributed feedback lasing in a novel family of rigid oligo-p- phenylene vinylene molecules. These show a remarkably low laser threshold for organic laser molecules doped into a passive polymer matrix and also amongst the best reported operational stability. It is shown that the rigid structure of the molecules allow a systematic tuning of the laser emission across the visible spectrum simply by increasing the length of the molecule, and it is argued that this rigid structure also needs to brood operational stability under laser action.

There are relatively few studies of systematic chemical design of organic laser materials, and so this study will be of significant interest to researchers in the field, and influence the design of future low threshold organic laser materials. The methodology used in the paper is appropriate for this kind of experiments and the data is of high quality. There are whether a number of areas in the manuscript which should be improved prior to the work being suitable for publication in Nature Communications.

The manuscript by Bona et al. contains an interesting report on a multidisciplinary investigation of certain zig-zag type polycyclical aromatic molecules (nanographenes) which exhibit amplified spontaneous emission (ASE) and which can be used – when dispersed into polystyrene- as active media in optically pumped distributed feedback (DFB) lasers.

Albeit the different aspects of the manuscripts are not completely new, the manuscript is a very high quality piece of work bringing together a quite unique combination. It demonstrates for the first time the use of this class of materials in DFB-lasers. A particularly impressive result is the realization of lasers spanning a rather large range of wavelengths from 485 nm to 685 nm.

Validity: The paper is well written and it contains very thorough and technically sound work. The group of authors brings together a long standing expertise ranging from the synthesis to device engineering and characterization. The last author has done pioneering work on different laser materials.

Originality and significance: The paper does neither report on a completely new material nor a completely new photophysics but it brings together a very interesting materials class with a rather sophisticated and clever device structure and a very thorough investigation of the optical gain behavior. The results are very significant and will inspire many follow up activities in the field of nanographenes.

Methodology: The data, the experimental procedures and the methodology to draw conclusions, the use of error bars, referencing of previous literature look very reasonable and convincing.

In summary, this is an important piece of science with high impact and I suggest to accept the paper after minor but mandatory revision.

I have a couple of minor comments on smaller mistakes, typos or unconvincing parts of the manuscript which need/should be addressed prior to publications of this nice manuscript:

Comment 1.1: In the affiliation part, the first “d” should be a “b”.

Response: *This has been corrected in the revised version (page 1, REV-1.1).*

Comment 1.2: On page 7, the authors are discussing the recent work by Morales-Vidales (actually with the same last author: “The emission spectrum covered by these COPVn ...” In here, it is somehow negatively discussed with a hint that six compounds were needed for a certain wavelength

range while only three compounds are needed for the nanographenes? I do not understand this. If it is only about the overall range, it is always only a discussion of TWO compounds, namely the most red-shifted and the most blue-shifted one. If it is about spectral tunability, the paper needs more data. The tunability of the individual compounds (Fig. 2b) seems to be rather limited.

Response: In this paragraph we attempted to condensate several ideas/concepts related to chemical aspects of the compounds. Seemingly, our explanations were not clear. In the comparison of FZn versus COPVn, we wished to highlight two main points of view:

1. *Oligomeric point of view.* With this, we mean that for instance from COPV1 to COPV2 in the COPVn series the oligomeric length has been increased by one unit, a phenyl-vinylene unit (8π electrons more in total from COPV1 to COPV2, see blue bonds in the scheme below); In the FZn series, from FZ1 to FZ2 the oligomeric length has been also increased by one unit, a part of an anthracene unit (8π electrons more in total, see scheme below with bonds in blue). So, the increase from one to another is very similar in term of total number of electrons, 8π electrons, but the consequences in the total electron absorption/emission shift is very different.

2. *π -electron delocalization point of view.* Related to the point above is the fact that in despite of the same number of 8π electrons more per monomer in the two series, the electronic disposition in one and another is on the origin of the differences. Namely, linear, 1D π -electron delocalization in the COPVn compounds and 2D π -electron delocalization the FZn (look that the added electrons in FZ2 contribute to the two molecular edges, the zig-zag and the arm-chair. This is an important aspect that has the beneficial effect of larger optical tuning/redshift per new π -electron and the detrimental effect of the activation of non-radiative processes. In the compromise of the two we importantly discover FZ3 as a near IR ASE emitter. This is one of the main strengths of the paper.

According to this discussion, the paragraph related to this discussion has been modified in the revised version (page 8, **REV-1.2a**):

Although our comment on page 7 about the comparison between COPVn and FZn referred to chemical aspects, we have also addressed another issue mentioned by the reviewer. Particularly the sentence “The tunability of the individual compounds (Fig. 2b) seems to be rather limited”)

As already mentioned in the original manuscript, the various lasers prepared for the various compounds were designed to emit close to the ASE wavelength, at which gain is maximum, and thus to minimize threshold. We had not explored the tunability range for a given compound. Since we think this is indeed an important point, we have prepared additional lasers with FZ2, emitting at shorter or longer wavelengths. The corresponding parameters for these devices have been included in Table 2 and the spectra for some of them in Figure 2b. According to the new data, the sentences of the original manuscript describing the emission range for a given compound have been modified in the revised version (pages 13 and 14, **REV-1.2b**). It should be noted that the labels of the devices based on FZ2 have been renumbered to assign letters A,B,C,...etc, as the grating period (and the emission wavelength) increases. Thus, the labels of FZ2 devices in the revised version have changed with respect to original manuscript.

Comment 1.3: On page 12, the authors claim that 700 nm are out of the “sensibility” (which I guess should read “sensitivity”) of the human eye. While the sensitivity goes down towards 780 nm, the general statement per se by the authors is wrong.

Response: *The reviewer is right. The word “sensibility” should be “sensitivity”. In addition, the eye sensitivity limit of the scotopic vision regime is 780 nm (after CIE 1951).*

Motivated by this comment, we tried to capture an image of the light emitted by a FZ3 laser by using a camera with more sensitivity than the one used to obtain the images for the other lasers. We found that the problem was not the sensitivity of the camera, but the fact that the laser intensity is not large enough (relative to noise, which is rather high due to the strong pump intensity needed to operate), to obtain an image with reasonable quality.

*According to this, in the revised manuscript we have changed the sentence that aims to justify why an image for a FZ3 laser is not shown (page 15, **REV-1.3**).*

Comment 1.4: The weakest part of the paper is the part “Discussion” which is more an extended conclusion as nothing really new is discussed but rather the statements from the results part of the paper are repeated. Besides the nice findings which are very interesting from an application oriented point of view, I personally would expect a little bit more of a discussion of the underlying photophysics or a discussion of future research directions.

Response: *In our manuscript, the discussions of the results have been done along with the results. We decided to do it in this way because it facilitates the reading and the comprehension. Taking into account that the various parts of the paper are about quite different aspects (Raman, ASE and DFB), we considered that leaving all the discussions for the “Discussion” section would difficult its understanding. Thus, in the “Discussion” section we have summarized the most important points of the different discussions already done along the paper. For example, some aspects of the photophysics of the compounds have been discussed in the section related to the low-T PL and Raman data. Nevertheless, a detailed understanding of the processes involved in the laser emission, as well as those responsible for the photodegradation will require further studies, such as transient absorption and pump-probe experiments. This is precisely one of the areas of future research interest what we intend to face in due course with the compounds investigated in this work.*

According to this reviewer suggestion and comments from the other reviewers, we have enlarged the discussions of different aspects (including the photophysics) in different locations of both, the Results and the Discussion sections. For example, some aspects of the photophysics in relation to the Raman data have been revised through the response to reviewer 3- comment 2.

With regards to including a discussion of future research lines, indeed, in the first submission just few aspects at this respect were mentioned. Thus, we have included a new paragraph to address this issue.

*Overall, and accordingly, the discussion section has been revised significantly (page 20, **REV-1.4**).*

Comment 1.5: In any case, I am expecting a more thorough discussion of the rather surprising stability results. Is there any idea about the underlying mechanisms which lead to such differences in the lifetimes, especially the extended one for FZ3?

Response: *The investigated nanographenes have shown a high stability, not only against photodegradation, but also against air and moisture (these two aspects had not been mentioned in the original submission and had also been discussed in the present version). This is mainly due to two situations acting together in benefit of the stability of these molecules: i) On one hand, their own*

chemical structures in which the most chemically reactive sites in the molecular peripheries are blocked with chemically inert and bulky aryl groups (kinetic blocking); and on the other, their inherent rigid molecular backbones which restrict excited state conformational mobility and non-radiative processes). And ii) the use of the compounds dispersed in a solid matrix (PS in this case), which, given the high diluted regime used, prevent intermolecular interactions and reactions, thus minimizing photoluminescence quenching and degradation. For FZ3, its excellent operational laser lifetime could be also ascribed to its rigid and largely delocalized backbone. Note that in the (few) previous papers related to the observation of ASE in nanographene molecules, very large stabilities were also observed. These were ascribed, after detailed photophysical studies (transient absorption and pump probe) to the robustness of the excited states of the nanographenes and the dispersion in a matrix.

According to this discussion, we made various changes in the revised manuscript:

- *In the results section related to the stability of the ASE (page 11, **REV-1.5a**).*
- *In the results section related to the operational laser lifetime of the devices (page 17, **REV-1.5b**).*
- *In the discussion section (page 18, **REV-1.5c**).*

Reviewers' comments and responses:

Reviewer #2 (Remarks to the Author):

The authors report nanographene-based lasers from three zigzag edged polycyclic aromatics. The detailed experimental are presented with proper explanations for their mechanisms. The authors demonstrate that the nanographene can efficiently provide absorption/emission wavelength bands, covering a large part of the visible spectrum, and approaching very much to the near-infrared region. These results and fabrication method could promote the application of graphene in exploring excellent laser devices. In my opinion, this manuscript is suitable for publication in Nature Communications.

More comments and suggestions are listed below.

Comment 2.1: In the manuscript, the absorption/emission wavelength bands of three nanographene molecules are studied. They show that the optical spectrum depends on the molecular size. In the three nanographene molecules, the FZ1 corresponds to the spectrum region with the shortest wavelength. Can nano-graphene further achieve shorter wavelengths? In addition, if we continue to get longer wavelengths by increasing the size of graphene, will we run into other constraints? What is the limitation of emission wavelength based on this mechanism? I suggest that the author discuss these issues in detail.

Response: *In the FZn (see scheme below) the smallest molecule should be anthracene, a well-known polycyclic aromatic hydrocarbon of the acene series which shows strong fluorescence and has been proposed for ASE lasing in previous studies. An alternative candidate is pyrene, and both can exhibit intense emission in UV region.*

The larger FZ4 in Scheme below should likely displays absorption and emission into the NIR region. The low energy lying absorption/emission should promote highly active non-radiative emission quenching decays which should eliminate ASE. In addition, the appearance of more developed/extended zig-zag edges would open new and/or more efficient degradation routes related with radical chemical reactions that would diminish very much the lifetime of the active materials.

The synthesis of FZ4 is still not done mainly due to synthetic challenges, but we are looking for alternative nanographenes with efficient emission in NIR region and hope for attaining NIR lasing.

*In the revised manuscript, we added a comment related to this issue (pages 18 & 19, **REV-2.1**):*

Comment 2.2: In the manuscript, it is mentioned that the film thickness, h (values for all the films listed in Table 1) were adjusted to ensure that the waveguides support only one transversal electric mode (TE_0). The transverse magnetic mode (TM_0) might also be supported for the film thickness given in Table 1. The author should check this statement.

Response: *The reviewer is right: all the films listed in Table 1 have, in addition to the TE_0 waveguide mode, also the TM_0 one. This can be predicted according to previous studies performed by some of us (see Calzado et al. Appl. Opt. 51, 3287, 2007; Ref. 31 of the manuscript). Nevertheless, we have checked this for the nanographene films, both, experimentally by the m -line technique*

(similarly as it was done in Ref. 31), and by modelling (using Hammer, M. 1-D mode solver for dielectric multilayer slab waveguides <http://www.computational-photonics.eu/oms.html>).

*According to this discussion we have modified a sentence in the revised version (Page, 5, **REV-2.2**):*

Reviewers' comments and responses:

Reviewer #3 (Remarks to the Author):

The authors described the demonstration of solution-processed nanographene distributed feedback lasers with the narrow single mode emission in the broad spectral range by using passive polymer films and resonators. The obtained low threshold laser devices are flexible and low-cost, which is attractive for practical applications. Meanwhile, these devices behave the long lifetimes and low thresholds. Overall, this work is quite interesting and potentially important for the development of practical flexible lasers. Thus, I would like to consider its publication after addressing the following concerns.

Comment 3.1: The spectral narrowing data of lasers should be provided in the main text or Supplementary. Meanwhile, the cavity mode of the resonant cavity should be measured or further discussed. It will be better if the authors show the linewidth narrowing of the cavity mode above the threshold rather than the narrowing of the spontaneous emission band.

Response: *As suggested by the reviewer, we made a new Figure to show the spectral narrowing of some of the lasers. This is Figure 3a in the revised version. The Figure 3a of the original manuscript (the one showing the output intensity versus the pump intensity) has been moved to the supporting information as Supplementary Figure 4, modified according to comment 3 of this same reviewer (see below).*

*A sentence in the text related to these figures has been included (page 15, **REV-3.1a**).*

*In relation to the comment about the cavity mode, we have introduced a new paragraph explaining the mechanisms involved in the lasing process in this type of lasers (page 13, **REV-3.1b**). Additionally, we have added a new figure (Figure 3b in the revised manuscript) showing the evolution of the emission spectrum as E_{pump} is increased. Several sentences related to this Figure have also been included (page 16, **REV-3.1b**). The Figure 3b of the original manuscript becomes Figure 3c in the revised version.*

Comment 3.2: For the discussion on the transitions in Figure 1c, some of vibronic progressions do not agree well with the measured Raman shifts quantitatively. Thus, the authors may not directly correspond these vibronic progressions with the observed Raman features since not all vibrational modes can be reflected by Raman scattering data.

Response: *We agree with the referee that not all vibrational modes in the Raman spectrum correspond with all vibronic modes. The selection rules from one and other processes are different. However, we have selected/settled the conditions for the Raman measurements in such a way that laser excitation of the Raman experiments were in resonance, near-resonance or pre-resonance conditions with the main absorption bands (otherwise responsible of the subsequent light emission and ASE). In these conditions the Raman intensity of the compounds is mostly dictated by the term A of the overall Raman intensity equation which corresponds with the so-called vibronic term of the Raman intensity which includes the “pure” electronic contribution by means of the transition dipole moment operator. On the other hand, the vibronic activity in the absorption/emission spectra is also controlled by the same quantum mechanical quantity (dipole transition moment). So by closer inspection of the nature of the Raman intensity in resonance and pre-resonance conditions and of the absorption and emission intensities, some correlations can be established, given that quantities with the same nature or origin govern both, vibronic and Raman mechanisms. This is the framework of our connection between vibronic frequencies and Raman frequencies. According to this discussion, we select the relevant frequency bands in the Raman spectra with the argument that these should be those bands dominating the spectrum in their own regions. With these assumptions, we did the*

correlations shown in the manuscript. Obviously, as the referee states, not all bands in the Raman spectrum can be recognized in the vibronic progressions but in our conditions, it is possible to do this if the appropriate conditions are settled. Finally, although the frequencies of the vibronic and Raman bands do not coincide exactly, in our comparisons in the paper the agreement is qualitative.

Based on this discussion we made several changes in the paragraphs related to the Raman results (pages 8 and 9, **REV-3.2**).

In addition, with regards to the resonant Raman discussion, a new figure has been included in the supporting information (Supplementary Figure 1b) showing Raman spectra in pre-resonance conditions. A brief discussion of this figure is included in the supporting information file (pages 1,2, **REV-3.2-SI**). Supplementary Figure 1 in the original manuscript becomes Supplementary Figure 1a in the revised version.

Comment 3.3: For Figure 3a, to visualize the threshold kink, it will be better to show the typical Log-Log plot instead of the used Log-Linear plot. The measured output intensity is collected from the full emission range or the ASE spectral range?

Response: Figure 3a of the original manuscript has been plotted in a log-log scale as suggested by the reviewer. The new figure has been moved to the supporting information file as Supplementary Figure 4. The measured output intensity is collected at the wavelength at which lasing is observed for each case. This is indicated in the caption of this Figure (page 5 of the supporting information file, **REV-3.3-SI**).

Comment 3.4: Does the polarization of the laser device vary with that of the pump laser beam or only along the direction of the grating lines no matter which incident polarization of the pump beam is used? Such information is helpful to determine the origin of the lasing polarization.

Response: We appreciate very much for this particular comment. In the original work we only focussed on laser modes associated to the TE_0 waveguide mode (whose light is polarized parallel to the grating lines). This is because from our experience with organic DFB lasers based on other materials and laser resonator architectures, the observed laser modes generally correspond to TE waveguide modes. Although in some of those cases we could observe lasing associated to TM modes, their thresholds were significantly larger. The reason why in this case it appears to be different is probably related to the use of the top layer resonator configuration. Nevertheless, further studies to clarify this issue are needed, which are out of the scope of the present work, whose main focus is in the use of nanographenes as active laser media.

In the original manuscript, the sentence about the polarization was rather confusing and incomplete. We apologize for that. The polarization properties of the emitted laser light for all the devices prepared has now been analysed in detail. We found that the lasers show either one or two peaks, which correspond to laser modes associated to a given waveguide mode (TE_0 or TM_0). The association can be easily made according to the polarization of the emitted laser peak: to TE_0 , when its light is polarized in a direction parallel to the grating lines, or to TM_0 when it is perpendicular to the grating lines. And this is independent on the polarization of the pump beam. Interestingly, the observation of either one or two peaks depend on the proximity of their emission wavelengths to the ASE wavelength.

According to this discussion we added several sentences in the revised version (pages 14 & 15, **REV-3.4**). We also added information in Table 2 and Fig. 2a to indicate, for each of the DFB lasers

prepared, the waveguide mode associated to each of the laser peaks observed. For the lasers in which two peaks are observed, data for both are shown.

Comment 3.5: The amplified spontaneous emission of the nanographene doped polymer film is linearly polarized, partially polarized or unpolarized?

Response: *We have measured the polarization of the ASE light collected from the edge of the films and no evidence of preferential polarization in a given direction was found. Taking into account that all the films prepared in this work support two waveguide modes, TE_0 and TM_0 , whose polarizations are parallel and perpendicular to the film edge, respectively, this result indicates that light traveling in both modes, experience gain due to ASE.*

*According to this, a sentence has been added in the revised manuscript (page 7, **REV. 3.5**).*

Finally, we comment some additional changes made:

- Various new references (from 48 to 58) have been added to support the discussions included. Subsequent references were renumbered accordingly.
- We have added the “Data availability” section, as requested by the editorial. Related to the data sources, we included proper sentences in the Figures.
- The “Lasing reporting document” requested upon submission has been updated, according to the revised version of the manuscript.

REVIEWERS' COMMENTS:

Reviewer #1 (Remarks to the Author):

The authors have carefully addressed the criticisms raised by the different reviewers. I support publication of the paper in the present form.

Reviewer #2 (Remarks to the Author):

The authors have addressed my concerns. In addition, the revised manuscript has improved the quality at these points that were mentioned by other referees. In my opinion, the revised version of the manuscript can be considered for publication in Nature Communication.

Reviewer #3 (Remarks to the Author):

The manuscript has been improved in view of the refined explanation and discussion. All my comments have been considered and replied in detail. The response to the previously raised concerns is reasonable and they have made careful efforts to provide a better presentation. To maintain the comprehensive illustration of lasing characteristics, the authors may consider moving the Supplementary Figure 4 into the Figure 3 in the main text. Overall, I consider that the current manuscript provides new insights on polymer-based lasing devices and thus is acceptable for publication on Nature Communications.

REVIEWERS' COMMENTS AND RESPONSES:

Reviewer #1:

The authors have carefully addressed the criticisms raised by the different reviewers. I support publication of the paper in the present form.

Reviewer #2:

The authors have addressed my concerns. In addition, the revised manuscript has improved the quality at these points that were mentioned by other referees. In my opinion, the revised version of the manuscript can be considered for publication in Nature Communication.

Reviewer #3:

The manuscript has been improved in view of the refined explanation and discussion. All my comments have been considered and replied in detail. The response to the previously raised concerns is reasonable and they have made careful efforts to provide a better presentation. To maintain the comprehensive illustration of lasing characteristics, the authors may consider moving the Supplementary Figure 4 into the Figure 3 in the main text. Overall, I consider that the current manuscript provides new insights on polymer-based lasing devices and thus is acceptable for publication on Nature Communications.

RESPONSE:

We thank very much for the reviewer's comments. We truly appreciate the time dedicated to read our work. Overall, thanks to their suggestions through all the reviewing process of the manuscript, it has improved significantly.

The only suggested change in this second revision is the one of reviewer 3 in relation to Supplementary Figure 4. According to his/her comment, this figure has been moved into the main text as Figure 3b. Consequently, former Figures 3b and 3c have been renamed as Figures 3c and 3d, respectively.